# High-Throughput Prediction of the Band Gaps of van der Waals Heterostructures via Machine Learning

**DOI:** 10.3390/nano12132301

**Published:** 2022-07-04

**Authors:** Rui Hu, Wen Lei, Hongmei Yuan, Shihao Han, Huijun Liu

**Affiliations:** Key Laboratory of Artificial Micro- and Nano-Structures of Ministry of Education and School of Physics and Technology, Wuhan University, Wuhan 430072, China; ruihu16@whu.edu.cn (R.H.); leiwen64@whu.edu.cn (W.L.); hmyuan@whu.edu.cn (H.Y.); hansh123@whu.edu.cn (S.H.)

**Keywords:** machine learning, van der Waals heterostructures, band gap, high-throughput screening

## Abstract

Van der Waals heterostructures offer an additional degree of freedom to tailor the electronic structure of two-dimensional materials, especially for the band-gap tuning that leads to various applications such as thermoelectric and optoelectronic conversions. In general, the electronic gap of a given system can be accurately predicted by using first-principles calculations, which is, however, restricted to a small unit cell. Here, we adopt a machine-learning algorithm to propose a physically intuitive descriptor by which the band gap of any heterostructures can be readily obtained, using group III, IV, and V elements as examples of the constituent atoms. The strong predictive power of our approach is demonstrated by high Pearson correlation coefficient for both the training (292 entries) and testing data (33 entries). By utilizing such a descriptor, which contains only four fundamental properties of the constituent atoms, we have rapidly predicted the gaps of 7140 possible heterostructures that agree well with first-principles results for randomly selected candidates.

## 1. Introduction

With the rapid and successful development in the study of two-dimensional (2D) materials [1,2,3,4,5], there has been a growing interest in the van der Waals heterostructures (vdWHs) due to their unique structures and rich physical properties [6,7,8]. Analogous to building with Legos, a vdWH can be viewed as stacking different 2D systems on top of each other. In fact, the weak interlayer interaction permits the superposition of virtually any given pair of 2D materials, such as graphene/phosphorene [9], silicene/graphene [10], phosphorene/SnX_2_ (X = S, Se) [11], graphene/hexagonal boron nitride [12], WSe_2_/Bi_2_Te_3_ [13], and so on. On the experimental side, the vdWHs can be prepared by using either bottom-up or top-down approaches [14,15,16]. The former makes use of successive deposition techniques, while the latter requires the fabrication of individual layers first and subsequent peeling off and assembly [7].

The emergence of vdWHs offers additional degree of freedom to tailor the electronic structure of 2D materials [17,18], especially for the band-gap tuning that leads to various applications such as thermoelectric [19,20] and optoelectronic conversions [21,22]. In general, the electronic band gap of a given vdWH can be predicted by using first-principles calculations [23,24,25]. Although the conventional local density approximation (LDA) [26] or generalized gradient approximation (GGA) [27] are relatively computationally efficient, they suffer from obvious band-gap underestimation. Accurate calculations require state-of-the-art techniques, such as the Heyd–Scuseria–Ernzerhof (HSE) hybrid functional [28] or the *GW* approximation of many-body effects [29]. However, both of them are computationally expensive and, thus, are restricted to small systems. As an alternative, the machine learning (ML) method has recently attracted considerable attention for band-gap prediction, which can efficiently manage a huge search space at an extremely low cost [30,31,32,33,34,35,36,37]. For example, by choosing 28 primary atomic properties as input features, Pilania et al. [30] obtained a kernel ridge regression (KRR) model to predict the band gaps of 1378 unique double perovskites, where the Pearson correlation coefficient can be as high as 97%. On top of the 136 compositional properties of inorganic solids, Zhuo et al. [31] proposed a support vector regression (SVR) model for gap prediction and found a small root mean square error (RMSE) of 0.45 eV. In addition, by leveraging 42 initial elemental features of chalcogenides, Wang et al. [33] developed a stacked ensemble learning (SEL)-gap model with a coefficient of determination (R^2^) value of 90%. It should be noted that the above-mentioned ML models usually contain a large number of input features, which is actually not beneficial for the high-throughput discovery of desired systems. Moreover, most of these models appear as black boxes, so a direct understanding of the underlying physics is quite necessary.

In this work, using group III, IV, and V elements as prototypical examples of constituent atoms, we adopt the SISSO (Sure Independence Screening and Sparsifying Operator) method [38,39] to propose a physically intuitive three-dimensional (3D) descriptor, by which the band gap of any vdWHs can be readily obtained. The strong predictive power of our descriptor is demonstrated by the good agreement between the SISSO-predicted gaps ($E_{g,\mathrm{pre}}$) and those calculated using accurate HSE functional ($E_{g,\mathrm{cal}}$), either inside or beyond the training data. As the input features only contain four fundamental properties of the constituent atoms, the 3D descriptor is very beneficial for the accelerated discovery of vdWHs with the desired band gaps.

## 2. Methodology

To obtain reliable training data for ML, we have calculated the band gaps of 325 vdWHs by using first-principles pseudopotential method, as implemented in the Vienna ab-initio simulation package (VASP) [40]. The hybrid functional within the HSE scheme [28] is adopted to overcome the gap underestimation in standard density functional theory (DFT), and the vdW interaction is considered by using the DFT-D3 exchange functional [41]. The plane-wave cutoff energy is set as 450 eV, and a 19 × 19 × 1 Monkhorst-Pack ***k***-mesh is sampled in the Brillouin zone. The energy convergence threshold is 10^−6^ eV, and the relaxed structure is determined until the residual force on each atom is less than 0.01 eV Å^−1^.

Based on the 325 samples in the training data, we adopt the SISSO approach [38,39] to obtain an optimized descriptor for predicting the band gap. Here, the input features include only four fundamental properties of constituent atoms, i.e., the atomic number Z, the Pauling electronegativity χ, the number of valence electrons VE, and the atomic radius r. A combination of algebraic operations is then recursively performed to extend the feature space, as defined by $H^{\left(m\right)}\equiv \left\{I,+,-,\;\times ,\;\;/,\;exp,\;log,\;\left|-\right|,\;\;\sqrt{\;},{\;}^{-1},\;{\;}^{2},\;{\;}^{3}\right\}$. Here, *m* means dimensional analysis, so that only meaningful combinations are allowed. By equipping the feature space with nonlinear operators in $H^{\left(m\right)}$, the intrinsically linear relation between observables and descriptor in the compressed sensing formalism is made nonlinear. At each iteration, $H^{\left(m\right)}$ operates on all possible combinations, and over 10^10^ features are constructed up to a complexity cutoff of 3. Such huge size can be effectively reduced by combining the sure independence screening (SIS) with the sparsifying operators (SO). Here, the SIS scores each feature with a metric and keeps only the top ranked. The subset extracted by the SIS is set to 80,000. After dimensionality reduction, the SO is used to pinpoint the optimal descriptor, which turns out to be physically interpretable. It should be mentioned that the SISSO algorithm demonstrates obvious advantages compared with other established ML approaches that suffer from huge and highly correlated feature spaces.

## 3. Results and Discussion

In the present work, we focus on the vdWH, which is composed of two graphene-like monolayers via vdW interactions along the out-of-plane direction. As illustrated in Figure 1, the system can be labeled by a nominal formula of AB/CD, where the A and B represent the two nonequivalent atoms in the upper layer, and the C and D represent those in the lower layer. Among these four atoms, the Pauling electronegativity of the A (C) atom is smaller than that of the B (D) atom by default. Considering the fact that vdWH permits the superposition of virtually any given pair of 2D materials [6], here we randomly select 26 graphene-like structures [42,43,44,45] as the constituent monolayers, including group V (N, P, As, Sb), group IV-IV (SiC, SiGe, SnSi, GeC, SnGe, SnC), and group III-V (BN, BP, BAs, BSb, AlN, AlP, AlAs, AlSb, GaN, GaP, GaAs, GaSb, InN, InP, InAs, InSb). For simplicity, only the conventional AA stacking pattern is considered, and the twist angle is 0°, which can in principle create $C_{26}^{2}$ = 325 different vdWHs (including 22 binary, 132 ternary, and 171 quaternary systems). Note that the vdW thickness is determined by minimizing the total energies of these heterostructures, where the vdW functional is explicitly considered in the DFT calculations. Besides, the stability and possible existence of 325 vdW heterostructures has been demonstrated in previously published work [43]. As there are only four atoms in the primitive cell, it is computationally ready to obtain the electronic band gaps of all these vdWHs from first-principles calculations, even with the high-level HSE scheme. The results are summarized in Appendix A, which will be used as the original dataset for SISSO training.

For any ML approaches, the selection of input features plays a crucial role in deriving the optimal model. In principle, one can adopt those related to the investigated system or the constituent atoms. Here, we consider the latter, since it can be readily obtained and is very beneficial for the high-throughput screening of the desired candidates. Among the 58 fundamental properties of the constituent atoms [46], we find that choosing the atomic number Z, the Pauling electronegativity χ, the number of valence electrons VE, and the atomic radius r could pinpoint an optimal descriptor that enables the accurate prediction of the band gap of the vdWHs. Table 1 lists these four kinds of input features for several constituent atoms from groups III, IV, and V. It should be mentioned that the 325 entries in the original dataset are randomly divided into 292 for effective training and 33 for real-time testing. Consequently, the SISSO-identified descriptor for gap prediction is given in a 3D form:(1)$$E_{g,\;\mathrm{pre}}=0.85\times D_{1}-25.32\times D_{2}-0.0003\times D_{3}-1.45$$
where $D_{1}$, $D_{2}$, and $D_{3}$ are defined as:(2)$$D_{1}=\frac{\left(VE_{A}+VE_{C})(\chi_{B}\times \chi_{D}\right)}{\left(r_{A}+r_{C})(\chi_{A}\times \chi_{C}\right)}$$
(3)$$D_{2}=|\frac{1}{VE_{B}(r_{A}+r_{D})}-\frac{1}{VE_{D}(r_{B}+r_{C})}|$$
(4)$$D_{3}=\frac{\left|Z_{A}-Z_{D}|+|Z_{B}-Z_{C}\right|}{\left(\chi_{B}{}^{3}+\chi_{D}{}^{3}\right)}$$

Figure 2 shows the intuitive linear correlation between the SISSO-predicted gaps ($E_{g,\;\mathrm{pre}}$) and those calculated by using HSE scheme ($E_{g,\;\mathrm{cal}}$). It is obvious that both the training (Figure 2a) and testing data (Figure 2b) are evenly distributed around the dashed line with slope one, suggesting the higher prediction accuracy of our SISSO descriptor. Indeed, the Pearson correlation coefficient is found to be 94% and 92% for the training and testing sets, respectively.

In addition to the strong predictive power discussed above, we should emphasize that the SISSO-derived descriptor is physically interpretable. Here, we focus on the first term (0.85 × $D_{1}$) of Equation (1), since it plays a major role [39] in the gap prediction. For the convenience of discussion, we rewrite Equation (2) as:(5)$$D_{1}=\frac{\left(VE_{A}+VE_{C}\right)}{\left(r_{A}+r_{C}\right)}\times \frac{\left(\chi_{B}\times \chi_{D}\right)}{\left(\chi_{A}\times \chi_{C}\right)}$$

That is, the SISSO-predicted gap is proportional to $\frac{\left(VE_{A}+VE_{C}\right)}{\left(r_{A}+r_{C}\right)}$ and $\frac{\left(\chi_{B}\times \chi_{D}\right)}{\left(\chi_{A}\times \chi_{C}\right)}$. For the $\frac{\left(VE_{A}+VE_{C}\right)}{\left(r_{A}+r_{C}\right)}$ term, if the A and C atoms are located in the same row of the periodic table, the number of valence electrons ($VE_{A}+VE_{C}$) increases, while the atomic radius ($r_{A}+r_{C}$) decreases with the increasing atomic number. As a consequence, the $\frac{\left(VE_{A}+VE_{C}\right)}{\left(r_{A}+r_{C}\right)}$ term increases, which leads to a larger band gap. This is reasonable since a decreased atomic radius ($r_{A}+r_{C}$) usually corresponds to a smaller bond length (or stronger bond strength), which would enhance band splitting, so a larger band gap can be expected. On the other hand, if the A and C atoms are in the same column of the periodic table, the number of valence electrons ($VE_{A}+VE_{C}$) keeps constant, while the atomic radius ($r_{A}+r_{C}$) increases with increasing atomic number. That is, we find a decreased $\frac{\left(VE_{A}+VE_{C}\right)}{\left(r_{A}+r_{C}\right)}$ and, thus, a smaller band gap, which can be understood by the same reason discussed above. For the $\frac{\left(\chi_{B}\times \chi_{D}\right)}{\left(\chi_{A}\times \chi_{C}\right)}$ term, since the Pauling electronegativities of the B and D atoms (anion-like) are relatively larger compared with those of the A and C atoms (cation-like), the significant difference between them would give rise to a larger band gap. Such a finding is consistent with the general belief that the band gap of an inorganic compound is approximately proportional to the electronegativity difference between the anions and cations [47,48].

Beyond the original dataset of 325 vdWHs, we have employed the SISSO-derived descriptor to predict the band gaps of a vast number of possible AB/CD heterostructures. Using group III, IV, and V elements as examples, it is assumed that A, B, C, and D can be chosen from 15 atoms: B, Al, Ga, In, Tl, C, Si, Ge, Sn, Pb, N, P, As, Sb, and Bi. As illustrated in Figure 3a, we can obtain a total of 7140 possible vdWHs, which includes 315 binary (calculated by $3\times C_{15}^{2}$), 2730 ternary ($6\times C_{15}^{3}$ = 2730), and 4095 quaternary systems ($3\times C_{15}^{4}$ = 4095). By leveraging the 3D descriptor given in Equation (1), we can quickly predict the band gaps ($E_{g,\;\mathrm{pre}}$) of all these 7140 vdWHs (including 325 entries in the original dataset). Figure 3b illustrates the distribution of $E_{g,\;\mathrm{pre}}$, where we see that most investigated heterostructures have smaller band gaps in the range of 0–1 eV. Besides, there are 786 systems exhibiting intermediate gaps from 1 to 3 eV, and 23 systems with gaps exceeding 3 eV. To further verify the predictive power of our 3D descriptor, we have calculated the band gaps of six vdWHs (BP/BiN, BSb/AsP, GeP/PP, SiAs/NN, AlC/AlAs, GeN/NN) by using an accurate first-principles approach (HSE scheme). Note that these vdWHs are randomly selected from Figure 3 and do not belong to the original dataset. As can be seen from Table 2, the band gaps obtained from the first-principles calculation ($E_{g,\;\mathrm{cal}}$) almost coincide with those predicted from the 3D descriptor ($E_{g,\;\mathrm{pre}}$). Once again, such an observation confirms the strong reliability of our SISSO approach, although only four fundamental properties of the constituent atoms are involved in the derived descriptor.

Before concluding our work, we should mention that the high-throughput results shown in Figure 3 also provide a very useful diagram to screen 2D functional materials with the desired band gap for potential applications, such as thermoelectric and optoelectronic conversions. Here, we focus on the thermoelectric materials that usually require a smaller gap (0–1 eV) to realize a relatively larger power factor. Among the 6331 possible candidates in Figure 3b, we choose the BSb/AsP vdWH as a checking example, where the SISSO-predicted gap is found to be 0.89 eV. Figure 4a plots the band structure of the BSb/AsP vdWH, as obtained from the first-principles calculations. We see that the conduction-band minimum (CBM) and the valence-band maximum (VBM) are both located at the **K** point, which gives a direct gap of 0.84 eV (HSE value) and is very close to the SISSO prediction. Based on the band structure, we can readily evaluate the Seebeck coefficient (S) by using the Boltzmann transport theory [49]. Note that S describes the ratio of voltage difference to the temperature gradient imposed on a given system and plays an important role in determining the thermoelectric-conversion efficiency that is usually evaluated by the dimensionless figure-of-merit $ZT\propto S^{2}$. Figure 4b shows the Seebeck coefficient of the BSb/AsP vdWH as a function of carrier concentration, where the result is the same along the armchair and zigzag directions. Remarkably, we observe that the S value at a typical-carrier concentration of 10^19^ cm^−3^ can reach 486 μV/K and 501 μV/K for the *n*- and *p*-type systems, respectively. Such values are significantly larger than those of state-of-the-art thermoelectric materials, suggesting a very promising thermoelectric performance by our vdWHs. If the carrier concentration is increased to 5 × 10^19^ and 1 × 10^20^ cm^−3^, we can, respectively, obtain an S value of ~300 μV/K and ~200 μV/K for both *n*- and *p*-type vdWHs, which are still comparable to those of many good thermoelectric materials.

## 4. Summary

In conclusion, our work offers a high-throughput solution to the fundamentally important issue of band-gap prediction, using the vdWHs as a prototypical class of examples. Unlike many ML algorithms, where the derived model usually appears as a black box, the SISSO approach employed in the present work can pinpoint a physically intuitive 3D descriptor that requires only four fundamental properties of the constituent atoms as input features. By leveraging such a simple-yet-efficient data-driven descriptor, one can deal with a huge search space (even for those systems with constituent atoms beyond Table 1) to screen promising candidates with the desired electronic band gap for potential applications, such as thermoelectric and optoelectronic conversions. It should be also noted that the SISSO approach established here can be adopted to predict the optical gap and entire band structure of vdWHs, as long as the corresponding training set is available.

## Figures and Tables

**Figure 1 nanomaterials-12-02301-f001:**
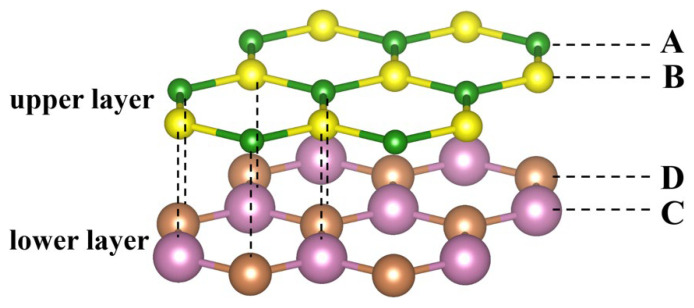
The crystal structure of AB/CD vdWH, where A, B, C, and D represent group III (B, Al, Ga, In, and Ta), group IV (C, Si, Ge, Sn, and Pb), and group V elements (N, P, As, Sb, and Bi).

**Figure 2 nanomaterials-12-02301-f002:**
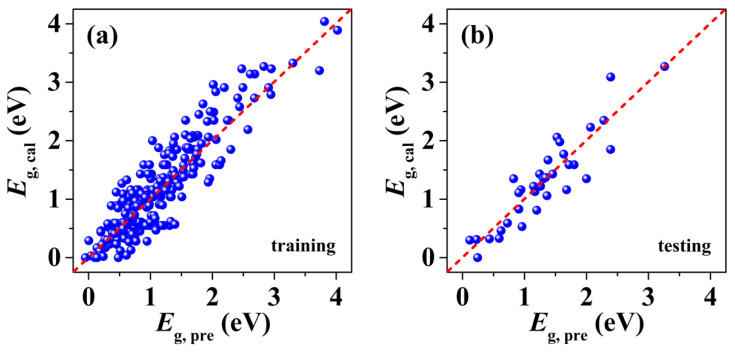
The intuitive linear correlation between the SISSO-predicted band gaps and those calculated by first-principles for vdWHs in the (**a**) training data and (**b**) testing data. The red dashed line represents equality.

**Figure 3 nanomaterials-12-02301-f003:**
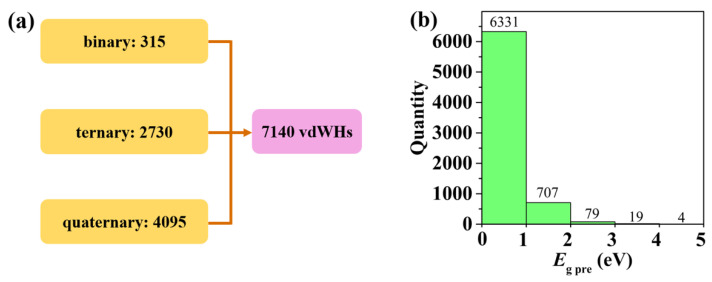
(**a**) Schematic illustration of 7140 AB/CD vdWHs, which includes 315 binary, 2730 ternary, and 4095 quaternary systems. (**b**) Distribution of 7140 vdWHs according to their SISSO-predicted band gaps.

**Figure 4 nanomaterials-12-02301-f004:**
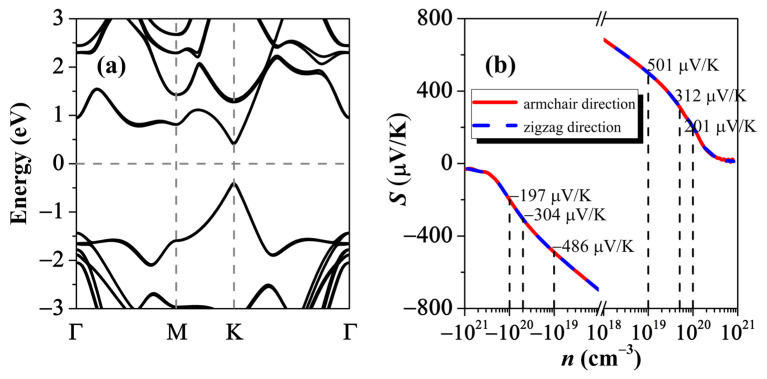
(**a**) The energy band structure, and (**b**) the Seebeck coefficient of the screened BSb/AsP vdWH.

**Table 1 nanomaterials-12-02301-t001:** The input features used for SISSO training, which includes the atomic number (Z), the Pauling electronegativity (χ, in units of eV), the number of valence electrons (VE), and the atomic radius (r, in units of Å) for group III (B, Al, Ga, In, and Ta), group IV (C, Si, Ge, Sn, and Pb), and group V elements (N, P, As, Sb, and Bi).

Elements	*Z*	*χ*	*VE*	*r*
B	5	2.04	3	0.95
Al	13	1.61	3	1.43
Ga	31	1.81	3	1.4
In	49	1.78	3	1.66
Tl	81	1.62	3	1.73
C	6	2.55	4	0.86
Si	14	1.98	4	1.34
Ge	32	2.01	4	1.4
Sn	50	1.96	4	1.58
Pb	82	2.33	4	1.75
N	7	3.04	5	0.8
P	15	2.19	5	1.3
As	33	2.18	5	1.5
Sb	51	2.05	5	1.6
Bi	83	2.02	5	1.7

**Table 2 nanomaterials-12-02301-t002:** Comparisons of the SISSO-predicted band gaps and those calculated by first-principles (HSE scheme), for six randomly selected vdWHs beyond the original dataset.

vdWHs	$E_{g,\;\mathrm{pre}}\;(\mathrm{eV})$	$E_{g,\;\mathrm{cal}}\;(\mathrm{eV})$
BP/BiN	0.35	0.38
BSb/AsP	0.89	0.84
GeP/PP	1.45	1.43
SiAs/NN	1.94	1.88
AlC/AlAs	2.10	2.09
GeN/NN	2.53	2.45

## Data Availability

The data presented in this study are available on request from the corresponding author.

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
