# Peer review of "High-Throughput Prediction of the Band Gaps of van der Waals Heterostructures via Machine Learning"

_nanomaterials, 2022, doi:10.3390/nano12132301_

Round 1
Reviewer 1 Report
There is a huge research effort in exploit machine and deep learning techniques for material design and characterization. Here the author interestingly use ML technique to predict the band gap of vdW structures instead of using other classical theoretical methods such as DFT, first-principles etc, with a considerable gain in terms of computational power and time.
However, a few points need to be addressed before considering publication
1. 2D vdW heterostructures possess usually high binding energy that leads to the formation at room temperature of stable exciton states. Therefore, a difference between electronic band gap (in the single particle approximation) and optical gap (related to exciton formation and recombination) must be considered. Is the method able to predict the optical or the electronic gap?
2. The ML based method provide just a number, i.e. the value of band gap. It does not provide nor the entire band gap diagram (function of Energy versus wavenumber), neither other information such as direct or indirect gap behavior. This is a strong limitation of the method. Can the author comment on that?
3. In the basic structure depicted in figure 1, the authors do not specify some structural parameter such as the distance between the two layer (vdW thickness), the lateral xy alignment and the relative twist angle. This latter has been proved as a key parameter to define the properties of vdW structures leading to the formation of several moiré structures (starting from the same layer constituents) with different properties. How did the author choose the vdW thickness, alignment and twist angle?
4. How does it work the algorithm? It perform an interpolation of the calculated gap with respect the four parameter in table one?
5. Where the formula (1) came from? Was this formula implemented in the algorithm? In this case, the whole method is just an analytical one.
Reviewer 2 Report
For a nonprofessional the method (SISSO) description sounds a bit cryptic. As it is an open access journal, perhaps the manuscript could benefit from :
1. A more clear description of how SISSCO is related to training.
2. (1) looks very simple and efficient. How was it derived from SISSO?
3. How can we be sure that a material is stable?
4. Please hint how Fig.4b was obtained? What is the figure of merit for materials with large S?
5. One sentence in the abstract could be improved: ... by using first-principles calculations, which is however restricted to
!that! ?calculations?
with small unit cell . Perhaps, restricted to small unit cells?
